# Molecular Composition of Serum Exosomes Could Discriminate Rectal Cancer Patients with Different Responses to Neoadjuvant Radiotherapy

**DOI:** 10.3390/cancers14040993

**Published:** 2022-02-16

**Authors:** Urszula Strybel, Lukasz Marczak, Marcin Zeman, Krzysztof Polanski, Łukasz Mielańczyk, Olesya Klymenko, Anna Samelak-Czajka, Paulina Jackowiak, Mateusz Smolarz, Mykola Chekan, Ewa Zembala-Nożyńska, Piotr Widlak, Monika Pietrowska, Anna Wojakowska

**Affiliations:** 1Department of Biomedical Proteomics, Institute of Bioorganic Chemistry Polish Academy of Sciences, 61-704 Poznan, Poland; ustrybel@ibch.poznan.pl (U.S.); lukasmar@ibch.poznan.pl (L.M.); asczajka@ibch.poznan.pl (A.S.-C.); paulinaj@ibch.poznan.pl (P.J.); 2Maria Sklodowska-Curie National Research Institute of Oncology, Gliwice Branch, 44-102 Gliwice, Poland; marcin.zeman@io.gliwice.pl (M.Z.); mateusz.smolarz@io.gliwice.pl (M.S.); chekan.mykola@gmail.com (M.C.); ewa.zembala-nozynska@io.gliwice.pl (E.Z.-N.); monika.pietrowska@io.gliwice.pl (M.P.); 3Wellcome Sanger Institute, Wellcome Genome Campus, Cambridge CB10 1SA, UK; kp9@sanger.ac.uk; 4Department of Histology and Cell Pathology, Faculty of Medical Sciences in Zabrze, Medical University of Silesia, 40-055 Katowice, Poland; lmielanczyk@sum.edu.pl (Ł.M.); oklymenko@sum.edu.pl (O.K.); 5Clinical Research Support Centre, Medical University of Gdańsk, 80-210 Gdańsk, Poland; piotr.widlak@gumed.edu.pl

**Keywords:** rectal cancer, small extracellular vesicles, exosomes, radiotherapy, plasma, metabolomics, proteomics

## Abstract

**Simple Summary:**

Exosomes could be used as biomarkers to predict and monitor the response to anti-cancer treatment, yet relevant knowledge is very limited in the case of rectal cancer. Here we applied a combined proteomic and metabolomic approach to reveal exosome components connected with different responses to neoadjuvant radiotherapy in this group of patients and processes associated with identified discriminatory molecules. Moreover, the composition of serum-derived exosomes and a whole plasma was analyzed in parallel to compare the biomarker potential of both specimens, which revealed the highest capacity of exosome proteome to discriminate samples of patients with different responses to radiotherapy.

**Abstract:**

Identification of biomarkers that could be used for the prediction of the response to neoadjuvant radiotherapy (neo-RT) in locally advanced rectal cancer remains a challenge addressed by different experimental approaches. Exosomes and other classes of extracellular vesicles circulating in patients’ blood represent a novel type of liquid biopsy and a source of cancer biomarkers. Here, we used a combined proteomic and metabolomic approach based on mass spectrometry techniques for studying the molecular components of exosomes isolated from the serum of rectal cancer patients with different responses to neo-RT. This allowed revealing several proteins and metabolites associated with common pathways relevant for the response of rectal cancer patients to neo-RT, including immune system response, complement activation cascade, platelet functions, metabolism of lipids, metabolism of glucose, and cancer-related signaling pathways. Moreover, the composition of serum-derived exosomes and a whole serum was analyzed in parallel to compare the biomarker potential of both specimens. Among proteins that the most properly discriminated good and poor responders were GPLD1 (AUC = 0.85, accuracy of 74%) identified in plasma as well as C8G (AUC = 0.91, accuracy 81%), SERPINF2 (AUC = 0.91, accuracy 79%) and CFHR3 (AUC = 0.90, accuracy 81%) identified in exosomes. We found that the proteome component of serum-derived exosomes has the highest capacity to discriminate samples of patients with different responses to neo-RT when compared to the whole plasma proteome and metabolome. We concluded that the molecular components of exosomes are associated with the response of rectal cancer patients to neo-RT and could be used for the prediction of such response.

## 1. Introduction

The first line of therapy for patients with locally advanced rectal cancer is total mesorectal excision supplemented with neoadjuvant treatment [1]. In the group of rectal cancer (RC) patients with a suspected increased risk of local recurrence or metastasis (i.e., T ≥ 3 or N+), it is advisable to use neoadjuvant radiotherapy (neo-RT) as a component of radical treatment [2]. Therapeutic response evaluation according to the tumor regression grading (TRG) system is essential for formulating treatment and survival forecasting [3], yet the actual prediction of tumor regression remains a challenge. Moreover, despite the benefits of preoperative neo-RT, generally leading to a reduction in tumor mass, such treatment may also result in radiation toxicity and other adverse effects [4]. Hence, a proper selection of patients who require a more aggressive preoperative treatment would be a desired component of tailored therapy. However, molecular markers which could be used for the prediction of efficacy and toxicity of neo-RT in locally advanced rectal cancer are still missed and searched for [5]. The rapidly developed omics technologies are widely used for searching cancer biomarkers that could be applied in monitoring the progression of disease and prediction of response to the treatment [6], which include proteomics and metabolomics [7,8,9,10]. A few studies revealed proteomic profiles of tissue and plasma of rectal cancer patients with different responses to RT [11,12,13]. On the other hand, few reports concerning metabolomic changes induced by the treatment of locally advanced rectal cancer patients were limited to the effects of neo-chemoradiotherapy [14,15]. Furthermore, a systemic approach combining both omics modalities was not applied in this field. 

Small extracellular vesicles (sEV), which include endosome-derived exosomes, carry many classes of bioactive molecules, including nucleic acids, proteins, lipids, and metabolites, which reflect the phenotype of parental cells [16]. Vesicles released by cancer cells (the so-called tumor-derived exosomes, TEX) and other cells present in the tumor microenvironment are key mediators in cell-to-cell communication involved in different aspects of cancer development, including growth, migration, angiogenesis, extracellular matrix (ECM) degradation, epithelial to mesenchymal transition (EMT), immune escape as well as resistance to the treatment. Therefore, numerous studies address exosomes (or sEV in general) as a potential source of cancer biomarkers with a particular focus on vesicles present in body fluids, which represent an emerging type of liquid biopsy [17,18]. Several reports confirm the important role of exosomes in the development of colorectal cancer, particularly their role in the reprogramming tumor microenvironment, immunomodulation, formation of the pre-metastatic niche, and drug resistance [18,19,20,21].

Transcriptomics and proteomics approaches have revealed potential exosomal biomarkers in colorectal cancer [22], yet knowledge about metabolomic and lipidomic components of such vesicles is very limited in colorectal cancer [23]. Recently, untargeted multi-omic analysis of exosomes, both present in serum and released by cell lines in vitro, revealed that the modulation of metabolism of fatty acids and amino acids is among characteristic features of colorectal cancer [24]. However, no data on radiotherapy-related changes of proteome and metabolome components of exosomes are available regarding this cancer. Nevertheless, it is generally assumed that radiation affects the molecular cargo of exosomes and these vesicles were implicated in the transmission of resistance to radiation as well as in the mediation of radiation-induced bystander effect [25,26]. Moreover, it is postulated that exosomes could be used as biomarkers for the prediction and monitoring of response to RT [27]. Here we aimed to address the hypothesis that molecular components of exosomes are associated with the response of rectal cancer patients to neo-RT and could be used for the prediction of such response. A combined proteomics and metabolomics approach based on different mass spectrometry techniques was applied to reveal relevant molecules and metabolic pathways. Moreover, the composition of serum-derived exosomes and a whole serum was analyzed in parallel to compare the biomarker potential of these vesicles.

## 2. Materials and Methods

### 2.1. Clinical Samples

Forty patients (15 females and 25 males, average age 66 years) treated radically due to adenocarcinoma located in the rectum (rectal cancer) were included in the study. All patients were given neo-RT in the total dose of 39–54 Gy (including 20 patients who obtained radiochemotherapy) that was completed 8 to 104 days (median 49 days) before surgery. Tumor regression grade (TRG; range 0–3) was assessed in the resected material based on the degree of fibrosis compared to the residual tumor: 0 (complete response, no residual tumor), 1 (<10% residual tumor), 2 (10–50% residual tumor) and 3 (>50% residual tumor). Seventeen patients were classified as “good responders” (radiosensitive tumors: TRG 0–1) and 23 patients were classified as “poor responders” (radioresistant tumors: TRG 2–3) to RT. The contribution of three applied neo-RT schemes (a total dose of 39, 42, and 54 Gy) was similar in both subgroups of patients. The disease status and clinicopathological information for all included patients are listed in Table 1. Blood samples were collected directly before surgery (the same day). The study was approved by a relevant Ethics Committee (local Ethics Committee at the National Research Institute of Oncology Branch in Gliwice, approval no. KB/430-50/12), and all blood donors provided an informed consent form indicating their conscious and voluntary participation.

### 2.2. Isolation and Characterization of Small Extracellular Vesicles (Exosomes)

Small EVs (total load) were isolated from the serum of patients with rectal cancer by the size exclusion chromatography (SEC) method adopted from Smolarz et al. [28] and Ludwig et al. [29] and optimized in our laboratory for MS-based analyses. The size and morphology of vesicles were evaluated by the dynamic light scattering (DLS) using the Zetasizer Nano-ZS90 instrument (Malvern, UK) and by transmission electron microscopy (TEM) using FEI Tecnai Spirit G2 BioTWIN at 120 kV acceleration, according to Thery et al. [30]. Known exosomal proteins, CD9, CD63, CD81, ALIX, TSG101 (primary antibodies: anti-CD63: (Thermo Fisher Scientific, Waltham, MA, USA ), 10628D, 1:1500; anti-CD9: (Santa Cruz Biotechnology, Dallas, TX, USA ), sc-13118, 1:500; anti-CD81: (Biorbyt, Cambridge, UK), orb388959, 1:500; anti-TSG101: (Becton Dickinson, Franklin Lakes, NJ, USA), 612697, 1:800; anti-Alix (Cell Signaling Technology, Danvers, MA, USA), 2171S, 1:1000, 2171S, 1:1000) were analyzed in sEV fraction by Western blot technique. The concentration of isolated PKH67-labeled sEV fraction was measured by Amnis ImageStream®X Mark II (Luminex, Seattle, WA, USA) flow cytometer. The concentration of proteins in the analyzed fraction was determined by the BCA method (PierceTM BCA Protein Assay kit, Thermo Fisher Scientific, Waltham, MA, USA), according to the manufacturer’s instructions.

### 2.3. Protein Extraction, Peptide Generation, and LC-MS/MS Analysis

Serum derived sEV were concentrated using Vivaspin 500 ultrafiltration tubes (100,000 MWCO, Sartorius, Göttingen, Germany) and 20 μL of the concentrate (corresponding to 15–20 μg of proteins) was mixed with 5 μL of 0.5% sodium deoxycholate (SDC). In the case of plasma samples, 2 μL of plasma was diluted with 118 μL of water, and next 10 μL of diluted plasma was transferred to new tubes and mixed with 100 μL of 1% sodium deoxycholate (SDC) in 50 mM NH_4_HCO_3_ buffer. Homogenized plasma and serum-derived sEV were centrifuged, transferred to new tubes, and digested with trypsin. The detailed protein extraction and digestion procedure is presented in Appendix A. The supernatant containing purified tryptic peptides was subjected to LC-MS/MS analysis. The label-free untargeted analyses were performed using a Dionex UltiMate 3000 RSLC nanoLC system coupled to a QExactive Orbitrap mass spectrometer (Thermo Fisher Scientific, Bremen, Germany), The obtained protein digests were separated on a C18 reverse-phase column using an acetonitrile gradient. All the raw data obtained for each dataset were imported into Protein Discoverer 2.1 package (Thermo Fisher Scientific) for protein identification and quantification. Protein identification was performed using the Swiss-Prot human database with a tolerance accuracy of 10 ppm for peptide masses and 0.08 Da for fragment ion masses. Protein was considered as identified if at least two peptides per protein were found by the search engine, and a peptide score reached the significance threshold FDR = 0.01 (assessed by the Percolator algorithm). The abundance of identified proteins was normalized to the total ion current (TIC). Detailed parameters of LC-MS/MS analysis are presented in Appendix A.

### 2.4. Metabolite Extraction, Derivatization, and GC/MS Analysis

25 uL aliquots of plasma samples were treated with 4 volumes of cold methanol. In the case of sEV, metabolites were extracted from obtained pellet subsequently with 200 uL of hexane, chloroform, methylene chloride, and finally methanol. Combined, dried extracts were derivatized with methoxyamine hydrochloride in pyridine and MSTFA. Detailed metabolite extraction and derivatization procedure is presented in Appendix A. Samples were subjected to GC/MS analysis directly after derivatization. Metabolites were separated and analyzed using the GC-MS system (TRACE 1310 GC oven with TSQ8000 triple quad MS from Thermo Scientific (Thermo Fisher Scientific, Rockford, IL, USA). Detailed parameters of GC-MS analysis are presented in Appendix A.

### 2.5. Lipid Extraction and Mass Spectrometry Analysis

Lipid separation was carried out according to MTBE extraction protocol [31], described in detail in Appendix A. Lipid profiling of plasma samples was performed using a Q-Exactive Orbitrap mass spectrometer (Thermo Fisher Scientific, Bremen, Germany) equipped with TriVersa NanoMate nanoflow ESI ion source (Advion BioSciences Ltd., Ithaca, NY, USA). Detailed parameters of shotgun mass spectrometry analysis are presented in Appendix A. MS data were processed using LipidXplorer software (ver. 1.2.8.1) developed at Max Planck Institute of Cell Biology and Genetics in Dresden (Germany) [32].

### 2.6. Statistical Analyses

Normalized data were log-transformed and scaled with the Pareto algorithm (mean-centered and divided by the square root of the standard deviation of each variable). Differences between independent samples were assessed using the T-test, Welch test, or U-Mann–Whitney test, dependent on the normality and homoscedasticity of data (assessed via the Shapiro–Wilk test and Levene test, respectively). In each case, the Benjamini–Hochberg protocol was used for the false discovery rate correction. However, due to the small sample size, none of the differences (except proteomic data from sEV samples) remained significant after the FDR correction. Therefore, the effect size analysis was employed to overcome this problem [33]. The Hedges’ g or the rank-biserial coefficient of correlation (an effect size equivalent of the U-Mann–Whitney test) was applied; the effect size ≥0.5 and ≥0.8 or ≥0.3 and ≥0.5 was considered medium and high, respectively [34]. Compounds with medium and high effect sizes were considered as differentiating the two groups of samples. Classical fold change or Hedges–Lehmann type fold change estimator was used for assessment pairwise ratios between the particular compound in the two groups. Principal component analysis (PCA) and hierarchical cluster analysis (HCA) based on the Euclidean distance method were performed to illustrate general similarities between samples. A single feature logistic regression classifier was constructed for each compound. Leave-one-out validation was performed, with several quality control metrics computed. The accuracy was computed as the mean of the TPR (true positive rate—sensitivity) and TNR (true negative rate—specificity) to be independent of group size. The entirety of the feature’s data was then used to create a ROC curve.

### 2.7. Bioinformatics Analyses

Proteomic data were analyzed using String ver.11.0—https://string-db.org/ (accessed on 15 October 2021) [35]. A list of genes corresponding to differentially expressed proteins (DEPs) was used to search for enriched Gene Ontology terms and Reactome pathways using hypergeometric testing with Benjamini–Hochberg multiple corrections. For predicting local network clusters (STRING) high confidence (0.7) was used as the minimum required interaction score. For clustering k-means clustering method was used. Metabolomic data were analyzed using MetaboAnalyst 5.0—https://www.metaboanalyst.ca/ (accessed on 22 October 2021). Metabolomic pathways associated with differentially accumulated metabolites (DAMs) were identified using the Metabolite Set Enrichment Analysis (MSEA), which is the metabolomic version of the Gene Set Enrichment Analysis (GSEA) approach. A list of selected metabolites with relative intensity was used as an input for the Quantitative Enrichment Analysis (QEA) algorithm, which was implemented using the hypergeometric test to evaluate the over-representation of a particular metabolite set; provided were fold-enrichment values and one-tailed *p*-values corrected for multiple testing. Lipidomic data analysis was performed using LION Lipid Ontology—http://www.lipidontology.com/ (accessed on 4 October 2021). The Ranking Mode tool was used for quantitative enrichment analysis of lipids associated with lipid pathways, functions, and organelle associations. Multi-omic data integration and analysis were performed using the Joint Pathway Analysis tool in MetaboAnalyst 5.0. For this purpose, a gene list corresponding to differentially expressed proteins and a list of differentially accumulated metabolites with fold changes were uploaded to integrated pathway analysis (based on the KEGG database). Enrichment analysis was performed using the hypergeometric test and for the integration method, loose integration by combining *p* values (with pathway-level weighted Z-test) approach was implemented. Moreover, DEPs and DAMs were subjected to integrated pathway analysis using the Reactome database. Over-representation analysis was performed for annotated DEPs and small molecules, using a binomial test with *p*-values corrected for the multiple testing (Benjamini–Hochberg procedure) [36]. Furthermore, the correlations between differentially expressed variables detected at several levels were defined by the Pearson coefficients; *p*-values < 0.05 were considered significant.

## 3. Results

### 3.1. Characteristics and Composition of Serum-Derived Exosomes 

Small extracellular vesicles (afterward called exosomes for simplicity) were isolated from serum by size exclusion chromatography and characterized according to relevant standards; the recovery of vesicles was about 1 × 10^10^/mL of serum. The morphology and size of vesicles revealed by transmission electron microscopy were comparable in both groups of patients (representative micrographs in Figure 1A). The size of vesicles estimated by the dynamic light scattering ranged between 30 and 100 nm (Figure 1B); noteworthy, a higher heterogeneity was observed in poor responders. The presence of exosome-specific markers (CD9, CD63, CD81, TSG101, and ALIX) was confirmed by Western blot in vesicles isolated from both groups of patients (representative samples in Figure 1C and the original Western blot in Appendix A). 

Proteins, metabolites, and lipids were identified and quantitated in isolated exosomes and corresponding samples of the whole plasma. Generally, the shotgun LC-MS/MS approach allowed the identification of 185 proteins in plasma samples and 279 proteins in exosomes (139 proteins overlapped between specimens; Appendix A). The complete list of 325 identified proteins is presented in Appendix A; molecular functions associated with these proteins are presented in Appendix A. An untargeted GC-MS-based approach allowed the identification of 110 and 50 metabolites in plasma and exosomes, respectively, of which 31 metabolites overlapped (Appendix A). The list of 129 small metabolites detected in both specimens is presented in Appendix A, major classes of detected metabolites are presented in Appendix A. It is noteworthy that the contribution of major classes of metabolites was different in both types of specimens: markedly fewer amino acids were detected in exosomes than in the whole plasma (2% vs. 20% of all detected metabolites), while fatty acids and lipids were markedly more abundant in exosomes than in the whole plasma (46% vs. 20% of all detected metabolites). Furthermore, a lipid profile of the whole plasma was analyzed by the shotgun LC-MS/MS approach, which revealed 452 lipid species (isomer groups) (Appendix A) that were annotated to 14 classes of lipids; the most numerous were triacylglycerols (TAGs), phosphatidylinositols (PIs), and triacylglycerols (DAGs) (Appendix A).

### 3.2. Proteins That Discriminated Patients with Different Responses to the Treatment

Among 325 proteins detected in analyzed specimens, several species showed significantly different abundance between good and poor responders (Appendix A). Due to the small sample size, none of the differences (except proteomic data from sEV samples) remained significant after the FDR correction. Therefore, compounds with a medium and high effect size were considered as differentiating the good and poor responders. There were 192 such differentially expressed proteins (DEPs) in exosome samples, which abundance was significantly different between both groups of patients. On the other hand, only 27 DEPs were detected in plasma samples (13 DEPs overlapped between both types of specimens). In further analyses, immunoglobulins were removed and the remaining DEPs coded by unique genes were used (Appendix A). Subsequently, there were 10 DEPs upregulated and 11 DEPs downregulated in plasma samples of good responders, 79 DEPs upregulated and 51 DEPs downregulated in exosomes of good responders were noted. Nine DEPs, namely C8G, ATRN, SERPINA4, PRDX2, GPLD1, CD5L, LGALS3BP, C1QA, and FCN3, were common for plasma and exosomes. Figure 2A shows normalized abundances of plasma DEPs showing the largest differences between groups (GPLD1, ATRN, APOC1, BCHE, APOF, F11), while Figure 2B shows normalized abundances of selected exosomal DEPs. All of the exosomal proteins presented in Figure 2B were significant after FDR correction.

To further characterize the potential of plasma and exosomal proteins to discriminate samples collected from both groups of patients, univariate classifiers were tested based on specific proteins. There were nine plasma proteins (4.9% of all detected proteins) for which a binary classification model (good responders vs. poor responders) performed with the receiver operating characteristics AUC higher than 0.7 (Appendix A). This included GPLD1 with AUC = 0.85 and accuracy of 74% (Figure 2C). On the other hand, there were 149 exosomal proteins (53% of all detected proteins) for which a single protein binary classification model performed with AUC higher than 0.7 (Appendix A). Among exosomal proteins that the most properly discriminated good responders and poor responders were C8G (AUC = 0.91, accuracy 81%), SERPINF2 (AUC = 0.91, accuracy 79%), CFHR3 (AUC = 0.90, accuracy 81%), THBS4 (AUC = 0.89, accuracy 81%), and HGFAC (AUC = 0.89, accuracy 78%) upregulated in exosomes of good responders (Figure 2D). Hence, one should conclude that the proteome component of serum-derived exosomes has a higher capacity to discriminate samples of patients with different responses to neo-RT when compared to the whole plasma proteome.

Functional enrichment analysis of detected DEPs was performed, which revealed several significantly overrepresented GO terms associated with DEPs present in exosomes (739 GO terms) and plasma (107 GO terms). Moreover, the analysis of functional interactions between DEPs was performed using the Reactome database. The TOP20 enriched processes and functions revealed by these analyses are shown in Appendix A. Overrepresented processes and functions associated with plasma DEPs were connected with response to stress, regulation of proteolysis, activation of the immune response, complement activation as well as lipoprotein and cholesterol metabolism. Similar processes and functions were associated with exosomal DEPs (except for vesicle-mediated transport, which was specific for this set of DEPs). To illustrate possible interactions among the 130 exosomal DEPs, functional network cluster analysis was performed (DEPs were divided into 5 clusters using the k-means clustering method). Figure 3 shows interactions between DEPs grouped into five functional clusters: immune response, vesicle-mediated transport, complement activation/protein activation, leukocyte mediated immunity/neutrophil degranulation, and cholesterol metabolism (the most significant GO terms connected with DEPs in individual clusters are presented in Appendix A).

### 3.3. Metabolites That Discriminated Patients with Different Responses to the Treatment

Among 129 metabolites detected in either plasma or exosomes, there were 22 and 23 differentially accumulated metabolites (DAMs), respectively, which levels were significantly different between poor responders and good responders (Appendix A). Two DAMs, namely pentadecanoic acid and sucrose, were common for both specimens, however, their exosome levels were higher in poor responders, while plasma levels were higher in good responders. The most numerous classes of DAMs were amino acids and sugars in plasma, while fatty acids and carboxylic acids in exosomes (Appendix A). Among exosomal DAMs, 12 were upregulated while 11 were downregulated in good responders. On the other hand, among plasma DAMs, 16 were upregulated, while 6 were downregulated in good responders. To further assess the ability of metabolites present in plasma and exosomes to discriminate samples of good responders and poor responders, univariate classifiers were tested based on specific compounds. There were 8 plasma DAMs (7% of all detected metabolites) for which binary classification models performed with AUC higher than 0.7 yet the accuracy of all of them was 50% (Appendix A). Similarly, there were 8 exosomal DAMs for which classification models performed with AUC higher than 0.7, yet only one of them showed high accuracy, namely 1,4-Dithiothreitol (AUC = 0.95, accuracy 75%) (Appendix A). Therefore, one could conclude similar potential of metabolites present in exosomes and plasma to discriminate patients with different responses to neo-RT, which was markedly lower than that of proteins present in exosomes.

Functional enrichment analysis of DAMs detected in plasma and exosomes was performed after annotation with their HMDB (Human Metabolome Database) IDs using the Quantitative Enrichment Analysis algorithm and The Small Molecule Pathway Database. Enriched pathways (*p*-value < 0.05) associated with DAMs are shown in Appendix A. Plasma DAMs were associated with lipids and amino acids metabolism. Metabolites upregulated in plasma of good responders were associated with the metabolism of glycerolipids, sphingolipids, and phosphatidylethanolamines, while metabolites upregulated in plasma of poor responders were associated with the metabolism of glutathione (Appendix A). On the other hand, metabolites upregulated in exosomes of poor responders were associated with energy metabolism (glycolysis, gluconeogenesis, trehalose degradation) and vitamin K metabolism (Appendix A). 

Analysis of lipidomic data revealed 108 differentially accumulated lipid (DALs) in patients’ plasma samples. This included 93 and 15 lipids upregulated in plasma of good and poor responders, respectively (the list of all DALs is presented in Appendix A). When univariate classification models were tested based on specific plasma lipids, 39 models showed AUC >0.70 (8.6% of all detected lipids), which included PE(34:2) (AUC = 0.81, accuracy 75%) and PS(34:5) (AUC = 0.79, accuracy 73%) (Appendix A). Hence, the hypothetical potential of plasma lipids and plasma proteins to discriminate patients with different responses to neo-RT was comparable. Functional enrichment analysis was performed by Lipid Ontology (LION) software using the “ranking mode” tool, then the significance of differential representation (good vs. poor responders) of terms related to lipid functions, cellular components, and lipid classification was estimated. Ten features, including mitochondrion, headgroup with a positive charge, glycerophosphoethanolamines, fatty acid with 16–18 carbons, and low lateral diffusion, were differentially represented between compared groups of patients (Appendix A). 

Levels of all detected small metabolites were used to perform unsupervised clustering of plasma and exosome samples. Both principal component analysis and hierarchical cluster analysis enabled a good separation of 40 plasma samples between good and poor responders (Appendix A). Even better separation of samples between groups was observed when a subset of exosome samples (*n* = 12) was analyzed (Appendix A). In marked contrast, however, unsupervised clustering of plasma or exosome samples did not allow the separation of two patients’ groups when proteomic or lipidomic datasets were analyzed (not shown). 

### 3.4. Integration of Data for Differently Expresses/Accumulated Proteins and Metabolites

To integrate proteomics and metabolomics datasets and reveal common pathways for DEPs and DAMs detected in plasma and exosomes of patients with rectal cancer who responded differently to neo-RT, Joint Pathway Analysis in MataboAnalyst 5.0 was performed. Figure 4A shows the KEGG pathways associated with plasma DEPs and DAMs that have the largest pathway significance (*p* < 0.05). The most significant pathways common for both classes of plasma components were complement/coagulation cascades (*p* = 3.04 × 10^−5^) and aminoacyl-tRNA biosynthesis (*p* = 1.39 × 10^−4^), as well as the metabolism of amino acids, fatty acids, and cholesterol. Figure 4B shows the KEGG pathways commonly associated with DEPs and DAMs detected in exosome samples. In this case, the most significant pathways included complement/coagulation cascades (*p* = 5.53 × 10^−33^) and staphylococcus aureus infection (*p* = 2.87 × 10^−12^) as well as platelet activation, cholesterol metabolism, ECM-receptor interaction, PPAR signaling pathway, focal adhesion, HIF-1 signaling, antigen processing/presentation, proteoglycans, and cell adhesion molecules (CAMs). 

In the next step, pathway enrichment analysis based on DEPs and DAMs found in plasma and exosomes was performed using the Reactome pathways analysis tool. TOP 20 significantly enriched Reactome pathways were presented in Figure 4C,D. In the case of plasma, significantly enriched pathways were connected with complement cascade (including ficolins and lectin pathways; *p*-value < 9.37 × 10^−4^), transport of fatty acids, digestion of dietary carbohydrate, free fatty acid receptors, synthesis of UDP-N-acetyl-glucosamine, and plasma lipoprotein assembly. In the case of exosomes, significantly enriched pathways were connected with the immune system (neutrophil degranulation and innate immune system; *p*-value < 4.28 × 10^−12^), complement activation cascade, and platelet functions (response to elevated platelet cytosolic Ca^2+^ and platelet degranulation). In the case of exosome components, a larger number of DEPs ad DAMs associated with each Reactome term was noted compared to plasma samples. The terms with the largest enrichment effect included the terminal pathway of complement and clotting cascade, as well as BRAF/RAF1 signaling and MAP2K/MAPK activation. 

Moreover, the possible relationships between differentiating components of plasma (21 DEPs, 22 DAMs, and 108 DALs) were addressed using Pearson’s correlation. The analysis revealed several correlations between the variables (Appendix A). Positive correlations between certain proteins and metabolites upregulated in plasma of good responders were noted, exemplified by ATRN correlated with stearic acid and isoleucine, or dodecanoic acid correlated with TAG(50:8), SM(33:1), PE(38:2), PI(40:3) or SM(31:1). On the other hand, APOE was negatively correlated with TAG(44:6) or PS(28:6) (details in Appendix A). Furthermore, differentiating components of exosomes (130 DEPs and 23 DAMs) were subjected to the same correlation analysis (Appendix A). Positive correlations between certain proteins and metabolites upregulated in exosomes of good responders were noted, exemplified by arachidonic acid, oleic acid, linoleic acid, cholesterol, and tocopherol correlated with CAMP, SLC2A14, IGFALS, HSPA8, F5, SFTPA2, ATRN, SERPINA6, PRDX2, PIGR, HLA-B, GPLD1, CD81, DBH, JUP, HRG, APMAP, and MPO. Moreover, positive correlations between glucose or 1,4-dithiothreitol and S100A8 or SLC2A1 upregulated in poor responders were noted (details in Appendix A).

## 4. Discussion

Exosomes can mediate response to radiotherapy by transferring proteins and other functional molecules to recipient cells and exerting different biological effects to modulate both radiosensitivity and the transmission of radioresistance [37]. However, there is no data on RT-related changes in proteome or metabolome cargo of exosomes in the case of rectal cancer. Here, a combined proteomic and metabolomic approach has been applied for the first time to study the molecular components of exosomes isolated from the serum of rectal cancer patients who responded differently to neo-RT.

Our primary observation was that the proteome component of serum-derived exosomes has a high capacity to discriminate samples of patients with different responses to neo-RT (much fewer DEPs were found if whole plasma samples were analyzed). Detected DEPs were functionally connected with activation of the immune response, complement activation, and platelet functions, as well as lipoprotein and cholesterol metabolism. A few DEPs have been previously reported as molecules connected with response to RT in colorectal cancer, including FGB, CD44, GLUT1/SLC2A1, PON1. It is noteworthy that increased levels of FGB observed in pre-treatment tissue biopsies were prognostic for poor response in rectal cancer patients subsequently treated with neoadjuvant chemo-radiotherapy [12]. The same study showed the association between a high level of proteins involved in platelet activation and blood coagulation, including TF and ACTB, in a group of poor responders. Hence, this is important to note that elevated expression of these proteins was observed here in serum exosomes of poor responders. Another DEP observed in our study, glucose transporter-1 (SLC2A1/GLUT1) that is critical in the metabolism of glucose, was over-accumulated in exosomes of poor responders (which correlated with the level of glucose in exosomes). A high GLUT1 expression was observed previously in radioresistant tumor cells, which was putatively associated with stimulation of hypoxia, and the regulation of different signaling pathways, such as MAPK and PI3K/AKT [38,39]. 

On the other hand, among DEP upregulated in exosomes of good responders was PON1, an important antioxidant enzyme, which elevated serum level was previously observed in rectal cancer patients in response to neoadjuvant radiochemotherapy [15]. Interestingly, signal transduction proteins S100A8 and S100A9 upregulated in exosomes of poor responders were previously proposed as CRC tumor-specific exosomal markers, involved in migration, leucocyte recruitment, tumor-promoting inflammation, and formation of premetastatic niches [18]. Though the potential to discriminate patients with different responses to RT was higher in the case of exosome components, there were nine DEPs common for plasma and exosomes. Two such DEPs, galectin-3-binding protein (LGALS3BP) and CD5 antigen-like (CD5L), significantly upregulated in both plasma and exosomes of poor responders, were connected with inflammatory response and immune surveillance, which are among the key processes upregulated in response to radiotherapy [40]. LGALS3BP has been reported to suppress colon inflammation and tumorigenesis through the downregulation of TAK1-NF-κB signaling [41]. Similarly, CD5L (a potential ligand for CD5) plays an important role in controlling the mechanisms of inflammatory responses [42].

Most studies on the role of exosomes in response to radiation and RT concern changes in transcriptome and proteome, yet much less is known about the metabolome component of these vesicles [43]. In general, only a few studies addressed RT-induced changes in metabolic profiles of rectal cancer [14,15,44], and there is no data about the potential of small molecules to discriminate patients who responded differently to radiotherapy. Here, we observed that metabolites present in exosomes and plasma had some potential to discriminate patients with different responses to neo-RT, and a few differentially accumulated metabolites (DAMs) were observed in both specimens (though DEPs detected in exosomes were much more numerous). DAMs upregulated in the plasma of good responders were associated with lipid and amino acid metabolism (note that DEPs upregulated in plasma of good responders were connected with lipid metabolism as well). Furthermore, several lipids (including different TAGs, PEs, PIs, SMs, and DAGs) had significantly different levels in the plasma of good and poor responders. Therefore, one should note that plasma components that showed significantly different levels between both groups of patients are generally associated with the metabolism of lipids. This is in agreement with papers documenting that radiotherapy resulted in disruption of plasma membranes [7] and induced changes in the level of lipids potentially connected with the inflammatory response [45]. In contrast, DAMs detected in exosomes were primarily associated with energy metabolism, including glycolysis and gluconeogenesis. For example, we observed significantly elevated levels of glucose in exosomes of poor responders. Noteworthy, an increased level of glucose and upregulation of glycolysis has been associated with a radioresistant phenotype (putatively via the induction of DNA repair pathways) [44].

The combination of proteomic and metabolomic datasets allowed us to reveal common pathways relevant for the response of rectal cancer patients to neo-RT. These processes included immune system response, complement activation cascade, platelet functions, metabolism of lipids, and cancer-related signaling pathways. Increasing evidence supports a role for complement in the development of cancer, and activity of complement system correlated with poor prognosis of colorectal cancer [46]. Similarly, platelets and platelets-derives sEV (putatively the most abundant EVs population in plasma) serve as regulators of cancer progression, and platelets-derives EV could promote proliferation and progression, crosstalk with the tumor microenvironment, and favor metastasis formation [47,48]. 

Exosomes carry enzymes and metabolites involved in the regulation of different aspects of cancer metabolism involved in response to radiation, including glycolysis, oxidative stress, and inflammation [49]. It is well documented that upregulation of glycolysis is associated with a radioresistant phenotype and exosomes serve as mediators of metabolic reprogramming in cancer cell response to RT [23,44]. This model was confirmed by our finding that glucose and phosphate accumulated in exosomes of poor responders. Similarly, two key enzymes involved in the metabolism of glucose, glucose transporter-1 (SLC2A1) glyceraldehyde-3-phosphate dehydrogenase (GAPDH), were also upregulated in exosomes of poor responders. Furthermore, we revealed at a proteomic and metabolomic level that mechanisms associated with response to RT are associated with the metabolism of lipids. Most of DEPs and DAMs upregulated in plasma good responders were connected with plasma lipoproteins, lipids, and cholesterol metabolism. On the other hand, exosomes of good responders were enriched in cholesterol and fatty acids, including PUFAs. PUFAs play an important role in cellular signaling, pro-inflammatory processes, and anti-oxidation as the reaction to radiation exposure [50]. Previous studies reported that extracellular vesicles are generally enriched in molecules involved in fatty acids transport and storage [51]. Paraoxonase-1 (PON1), upregulated in exosomes of good responders, is an important antioxidant enzyme linked to cellular mitochondria-associated membranes, and in high-density lipoproteins (HDL), protects the cell from oxidative stress [15]. On the other hand, two DEPs upregulated in exosomes of poor responders, fatty acid-binding protein 5 (FABP5) and CD5L, are also involved in lipid metabolism. CD5L, a key regulator of lipid synthesis, decreases the content of PUFAs and limits the expression of pro-inflammatory genes. FABP5 has been shown to deliver ligands to PPAR-β/δ in the nucleus and to increase angiogenesis through the PPAR-γ-VEGF signal transduction [52].

## 5. Conclusions

The Multi-omics approach applied in this study allowed us to reveal several proteins and metabolites, which levels in serum-derived exosomes discriminated patients with different responses to neo-RT. These molecules were associated with a few common pathways relevant to respond to the treatment, including the immune system, complement activation cascade, platelet functions, metabolism of lipids, and metabolism of glucose. Moreover, the highest number of molecules that had significantly different levels between good and poor responders was observed in the proteome component of exosomes, which suggested a high capacity of this particular fraction of blood to discriminate patients who differently responded to neo-RT. Hence, proteome components of serum-derived exosomes appeared a potential source of biomarkers for the prediction of response to neoadjuvant treatment in rectal cancer patients. Besides, the integration of metabolomic and proteomic data reveals novel insights into the role of exosomes in response to cancer treatment.

## Figures and Tables

**Figure 1 cancers-14-00993-f001:**
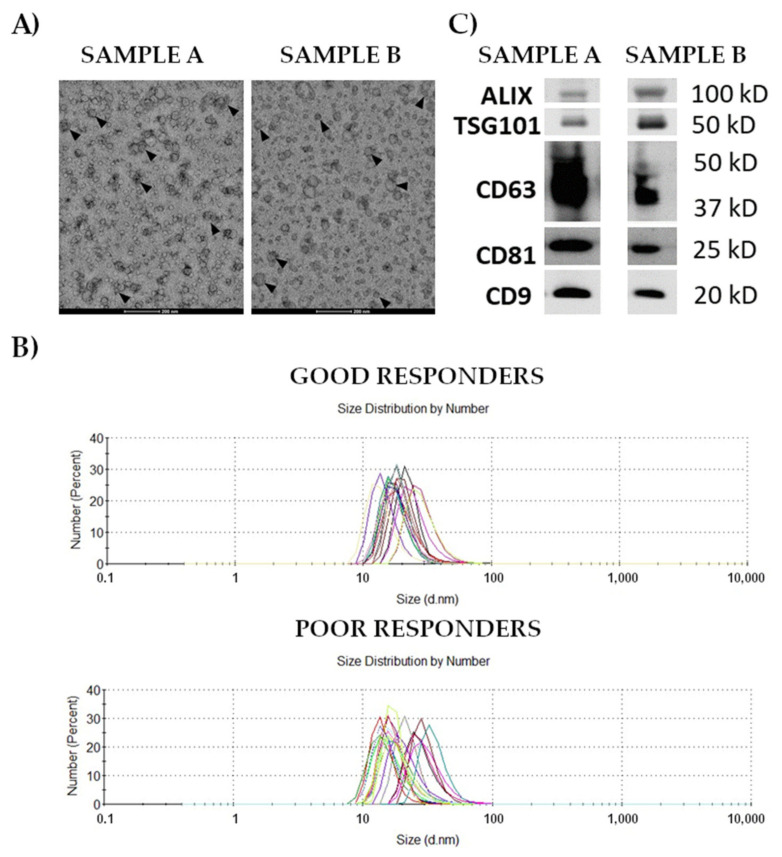
Characterization of exosomes isolated from the serum of patients with rectal cancer. Panel (**A**)—Morphology of vesicles analyzed by transmission electron microscopy at 87,000× magnification in samples representative for good and poor responders (samples A and B, respectively); exosomes are marked with arrows. Panel (**B**)—Size of vesicles estimated by dynamic light scattering in samples of good and poor responders. Panel (**C**)—Western blot analysis of exosomal markers in samples representative for good and poor responders A and B, respectively).

**Figure 2 cancers-14-00993-f002:**
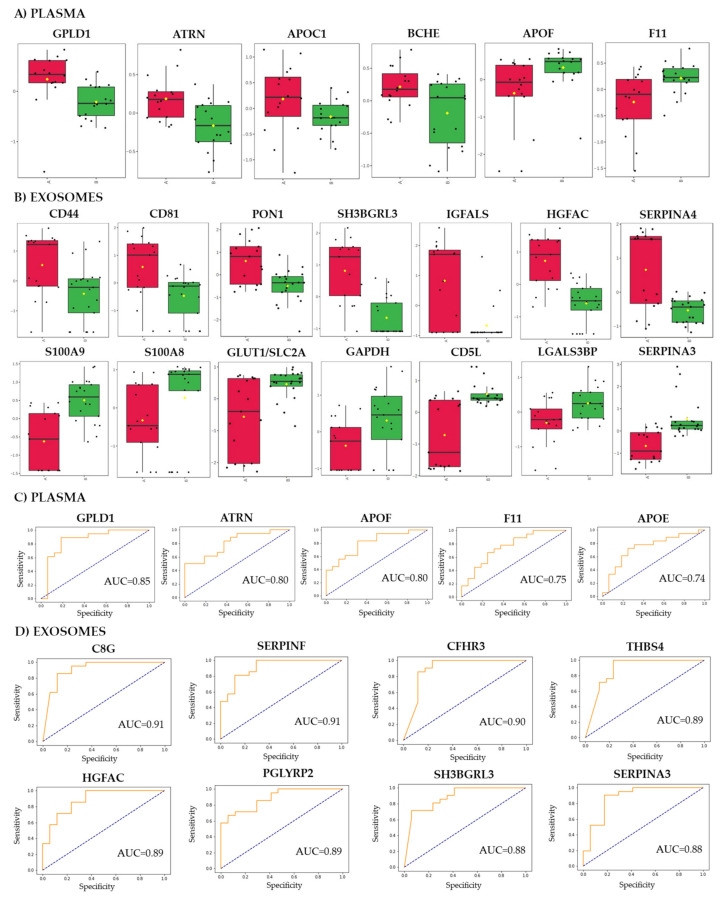
Differently expressed proteins in samples of rectal cancer patients with different responses to the treatment. The normalized levels of selected DEPs in plasma (Panel (**A**)) and exosomes (Panel (**B**)) in groups of good (marked in red) and poor (marked in green) responders. Boxplots show median, upper and lower quartile, maximum and minimum (yellow diamond indicated mean level). The performance of univariate classification models based on selected DEPs detected in plasma (Panel (**C**)) and exosomes (Panel (**D**)).

**Figure 3 cancers-14-00993-f003:**
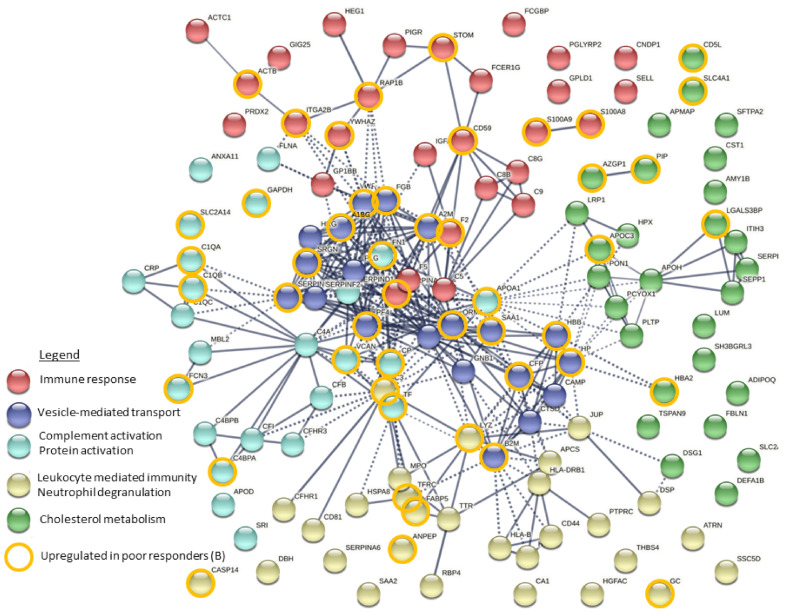
An interaction map of differentially expressed proteins detected in exosomes. Proteins that belong to five different clusters are color-coded; proteins upregulated in poor responders have orange borders.

**Figure 4 cancers-14-00993-f004:**
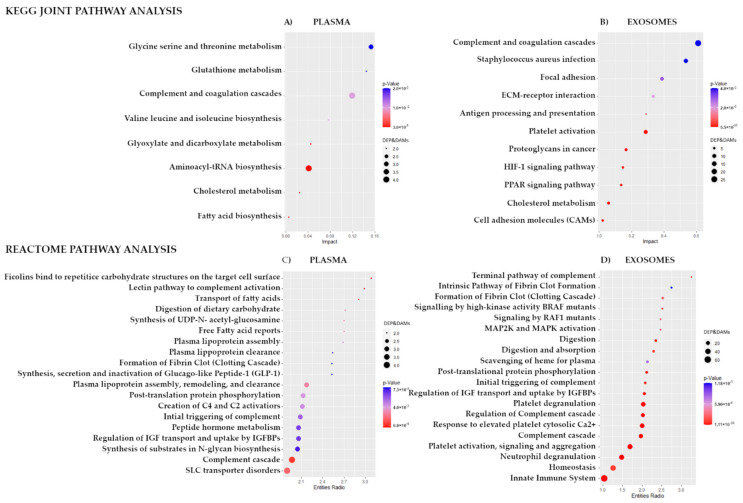
Pathways that were commonly associated with differentially expressed proteins and differentially accumulated metabolites. Statistically significant joint KEGG pathways that reflect the contribution of all DEPs and DAMs detected in plasma (Panel (**A**)) and exosomes (Panel (**B**)). TOP 20 significant Reactome pathways associated with DEPs and DAMs detected in plasma (Panel (**C**)) and exosomes (Panel (**D**)).

**Table 1 cancers-14-00993-t001:** Clinical characteristics of rectal cancer patients included in the study.

		Total *n* (%)	Good Responders *n* (%)	Poor Responders *n* (%)	Difference *p*-Value (Test)
Sex	Females	15 (37.5)	5 (29.4)	10 (43.5)	0.57 (Chi2)
	Males	25 (62.5)	12 (70.6)	13 (56.5)	
Age (years)	mean (S.D.) median	65.9 (9.8) 65.5	64.9 (12.2) 67	66.5 (7.8) 65	0.98 (M-W U)
BMI	mean (SD)	26.2 (3.5)	25.0 (3.5)	27.0 (3.3)	0.047 (M-W U)
Clinical Stage	II	13 (32.5)	5 (29.4)	8 (34.8)	1.0 (Fisher)
	III	25 (62.5)	11 (64.7)	14 (60.9)	
	IV	2 (5.0)	1 (5.9)	1 (4.3)	
RT scheme	39 Gy	17 (42.5)	8 (47.1)	9 (39.1)	0.1 (Fisher)
	42 Gy	16 (40.0)	4 (23.5)	12 (52.2)	
	54 Gy	7 (17.5)	5 (29.4)	2 (8.7)	
RT/CT		20	10	10	
Time RT/S (days)	mean (SD) median	52.7 (20.3) 49	54.6 (20.2) 52	51.3 (20.7) 49	0.53 (M-W U)
Surgery mode	AR	26 (65.0)	10 (58.8)	16 (69.6)	0.7 (Chi2)
	APR	14 (35.0)	7 (41.2)	7 (30.4)	
ypT	0–2	13 (32.5)	7 (41.2)	6 (26.1)	0.5 (Chi2)
	3	27 (67.5)	10 (58.8)	17 (73.9)	
ypN	negative	24 (60.0)	13 (76.5)	11 (47.8)	0.1 (Fisher)
	positive	16 (40.0)	4 (23.5)	12 (52.2)	
LNY	mean (SD)	12.3 (5.8)	12.5 (5.1)	12.1 (6.4)	0.59 (M-W U)

BMI, body mass index; RT, neoadjuvant radiotherapy; CT, chemotherapy; Time RT/S, the time from completion of RT to surgery; LNY, node yield; S.D., standard deviation; M-W U, Mann–Whitney U-test.

## Data Availability

The data presented in this study are available on request from the corresponding author. Moreover, the mass spectrometry proteomics data have been deposited to the ProteomeXchange Consortium via the PRIDE [53] partner repository with the dataset identifier PXD031556.

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
