# Peer review of "Molecular Composition of Serum Exosomes Could Discriminate Rectal Cancer Patients with Different Responses to Neoadjuvant Radiotherapy"

_cancers, 2022, doi:10.3390/cancers14040993_

Round 1

Reviewer 1 Report

Legend Figure2: "good and bad responder" - should include red and green

Author Response

Response to Reviewer 1 Comments

Point 1: Legend Figure2: "good and bad responder" - should include red and green

Response 1: The revised manuscript was corrected accordingly (lines 305-306).

Reviewer 2 Report

The current study investigates the potential of proteomics and metabolomics from serum extracellular vesicles (i.e. exosomes) and plasma to predict the response to neoadjuvant radiotherapy in rectal cancer. The study is highly relevant since patient outcome could highly differ between individuals. Analysis of proteins and metabolites present in serum and in serum exosomes could allow for a non-invasive prediction of the treatment response. The following aspects should be addressed to strengthen the conclusions of the study:

  1. Figure 1. TEM shows different types of structures, with different sizes and morphology. The authors should point with an arrow at representative sEVs in the micrography.
  2. Please add further information of the primary antibodies used for western blot (i.e. manufacturer, clone or catalogue number, dilution).
  3. To further support the findings of the proteomics study, differentially expressed proteins should be confirmed by western blotting of serum EVs.

Author Response

Response to Reviewer 2 Comments

Point 1: TEM shows different types of structures, with different sizes and morphology. The authors should point with an arrow at representative sEVs in the micrography.

Response 1: The revised manuscript was corrected accordingly. Arrows have been added in the revised Figure 1.

Point 2: Please add further information of the primary antibodies used for western blot (i.e. manufacturer, clone or catalogue number, dilution).

Response 2: The revised manuscript was corrected accordingly (lines 131-134).

Point 3: To further support the findings of the proteomics study, differentially expressed proteins should be confirmed by western blotting of serum EVs.

Response 3: Thank you. We obviously agree that for the further study proposed DEPs should be confirmed by other techniques, including WB. We are going to continue the studies to perform further validation of potential proteomic biomarkers. However, the assumption of these studies was to check whether combined proteomics and metabolomics approaches based on different mass spectrometry techniques will allow revealing relevant molecules and pathways connected with the response of rectal cancer patients to neo-RT.

Reviewer 3 Report

Overall, this is a carefully conducted study of general interest if made publically available (see comments below).

1. The Abstract is not informative as should contain some details on the markers identified (eg GPLD1 in plasma; C8G, SERPINF2 , CFHR3 etc in exosomes). 

2. Did overall EV content per ml after SEC differentiate poor and good responders as well?

3. While GO term analysis of the DEP plasma and exosome proteins is interesting, it is not very informative clinically. Here, it may be more important to evaluate combinations of DEPs, together with other clinical information such as TNM staging. This needs to be addressed.

4. In the Methods section the authors state that due to small sample size, none of the differences (except proteomic data from sEV samples) remained significant after FDR correction.  This should be clearly indicated in the Results section. Furthermore, it is important to state in the Result section which DEP(s) in exosomes was/were significant after FDR correction.

5. For studies like this, the data presented should be made publicly available as there are no understandable privacy issues in metabolomic or proteomic data sets of this kind. 

Author Response

Response to Reviewer 3 Comments

Point 1: The Abstract is not informative as should contain some details on the markers identified (eg GPLD1 in plasma; C8G, SERPINF2 , CFHR3 etc in exosomes).

Response 1: Thank you for your comment. The revised abstract was corrected accordingly (lines 39-42)

Point 2: Did overall EV content per ml after SEC differentiate poor and good responders as well?

Response 2: The concentration of EVs after SEC was measured only for a few chosen samples to determine the recovery of vesicles. We have measured protein concentration after SEC for all studied samples. Good and poor responders have similar variances, there were no differences in the mean protein concentrations between the patient groups which is confirmed by the Student's t-test (p-value = 0.316).

Point 3: While GO term analysis of the DEP plasma and exosome proteins is interesting, it is not very informative clinically. Here, it may be more important to evaluate combinations of DEPs, together with other clinical information such as TNM staging. This needs to be addressed.

Response 3: Correlation between DEPs and clinical data (LNY, ypN+ 0/1, ypT, cTNM, ypTNM) defined by Pearson coefficient was not significant.

Point 4: In the Methods section the authors state that due to the small sample size, none of the differences (except proteomic data from sEV samples) remained significant after FDR correction. This should be clearly indicated in the Results section. Furthermore, it is important to state in the Result section which DEP(s) in exosomes was/were significant after FDR correction.

Response 4: The revised manuscript was corrected accordingly (lines 272-275 and 287).

Point 5. For studies like this, the data presented should be made publicly available as there are no understandable privacy issues in metabolomic or proteomic data sets of this kind.

Response 5: The proteomic data are publicly available from the Pride repository with temporary identifier PXD031556 (lines 610-612). Reviewer account details:

Username: [email protected]

Password: 2rPrqvA1

Unfortunately, we were not able to publish metabolomic data so far, because the standard time for publication of metabolomic data in the MetaboLights repository is about 28 days: “Due to increased data submissions, the curation stage may be longer than our standard 28 days. We apologize for any inconvenience and please contact us if you have any specific requirements”.The Cancer journal does not require confirmation of making the data public during the manuscript submission process. However, for further studies, we will make the data public before manuscript submission. All data presented in this study are available on request from the corresponding author.

Round 2

Reviewer 3 Report

The authors have addressed all points raised except Point 3:

Point 3: While GO term analysis of the DEP plasma and exosome proteins is interesting, it is not very informative clinically. Here, it may be more important to evaluate combinations of DEPs, together with other clinical information such as TNM staging. This needs to be addressed.

Response 3: Correlation between DEPs and clinical data (LNY, ypN+ 0/1, ypT, cTNM, ypTNM) defined by Pearson coefficient was not significant.

The authors should perform a multivariate or at least univariate analysis that will include clinicopathological data.

Author Response

Response to Reviewer 3 Comments #2

Point 3: While GO term analysis of the DEP plasma and exosome proteins is interesting, it is not very informative clinically. Here, it may be more important to evaluate combinations of DEPs, together with other clinical information such as TNM staging. This needs to be addressed.

Response 3: Correlation between DEPs and clinical data (LNY, ypN+ 0/1, ypT, cTNM, ypTNM) defined by Pearson coefficient was not significant.

Point #2: The authors should perform a multivariate or at least univariate analysis that will include clinicopathological data.

Response #2: We performed a univariate analysis that included clinical data. Depending on clinical metadata traits two-sample statistical analysis or ANOVA/Welch, ANOVA/Kruskal-Wallis was applied based on the data's normality and homoscedasticity (assessed via the Shapiro-Wilk and Levene tests respectively).  The results of the analysis were not statistically significant. Please find attached the results of the analysis in zip file: Response#2Univariate.